# Learning Quadrupedal High-Speed Running on Uneven Terrain

**DOI:** 10.3390/biomimetics9010037

**Published:** 2024-01-05

**Authors:** Xinyu Han, Mingguo Zhao

**Affiliations:** 1Department of Automation, Tsinghua University, Beijing 100084, China; hanxy21@mails.tsinghua.edu.cn; 2Beijing Innovation Center for Future Chips, Tsinghua University, Beijing 100084, China

**Keywords:** reinforcement learning, quadrupedal robot, high-speed locomotion

## Abstract

Reinforcement learning (RL)-based controllers have been applied to the high-speed movement of quadruped robots on uneven terrains. The external disturbances increase as the robot moves faster on such terrains, affecting the stability of the robot. Many existing RL-based methods adopt higher control frequencies to respond quickly to the disturbance, which requires a significant computational cost. We propose a control framework that consists of an RL-based control policy updating at a low frequency and a model-based joint controller updating at a high frequency. Unlike previous methods, our policy outputs the control law for each joint, executed by the corresponding high-frequency joint controller to reduce the impact of external disturbances on the robot. We evaluated our method on various simulated terrains with height differences of up to 6 cm. We achieved a running motion of 1.8 m/s in the simulation using the Unitree A1 quadruped. The RL-based control policy updates at 50 Hz with a latency of 20 ms, while the model-based joint controller runs at 1000 Hz. The experimental results show that the proposed framework can overcome the latency caused by low-frequency updates, making it applicable for real-robot deployment.

## 1. Introduction

### 1.1. Background

The locomotion control of quadrupedal robots has become a research focus in recent years. Quadrupedal robots are believed to be capable of traversing more challenging terrains and performing more agile motions than wheeled robots. However, controlling a quadruped is also more challenging. Traditional model-based control contains algorithms such as model-predictive control [1,2] and whole-body control [3], which require manually decided structures and empirically tuned parameters for the best performance.

Reinforcement learning has been widely applied in locomotion control. Benefiting from the universal approximation ability of the neuron network policy model and the offline optimization of the reinforcement learning algorithm, the control policy optimized by reinforcement learning can capture complex robot dynamics. Reinforcement learning has been employed to control low-speed quadrupedal walking on uneven terrains [4,5,6,7], or high-speed running on flat ground [8,9]. RL is also implemented in bipedal locomotion controllers, resulting in a better performance regarding dynamic model uncertainty [10], fast yet robust maneuvers [11,12], and the precise control of stepping point [13] or gait [14].

Most of the existing RL-based control policy gives target joint or foot positions as output, which are tracked by a PD controller with fixed gains [6,15,16,17,18]. These methods introduce a constant impedance to the joints. Such an impedance will surpass the disturbance forces exerted by the ground to the robot. When the robot runs over uneven ground at high speed, the disturbance will severely decrease the stability. In Jin et al. [9], Choi et al. [19], the policy is updated with a high frequency and low latency to respond to such disturbances in a timely manner. In these methods, the control policies applied in agile locomotion control have an update rate of from 100 to 500 Hz and a latency of less than 10 ms.

### 1.2. Motivation

In contrast to legged robots, their animal counterparts cannot respond to external stimulation in such a short time. Nevertheless, they demonstrate a fast locomotion ability on complex natural terrains. When animals move a joint, a group of antagonistic muscles driving that joint will contract simultaneously [20]. As the contraction of these muscles will change their force–length relation and force–velocity relation [21], the stiffness and damping factor of the joint will change during motion. Having joints with adjustable impedance has advantages in resisting different types of disturbances. As qualitative examples, a leg following a swinging trajectory with low joint impedance will pass little disturbance force onto the torso if it collides with an obstacle. Supporting legs with high-impedance joints can stabilize the torso when they encounter external pushing forces.

Inspired by the variable impedance joints of animals, we designed a control framework consisting of a state estimator, a reference trajectory generator, an RL-based control policy with a low update rate, and joint controllers running at a high frequency. In this framework, the reference trajectory generator provides target joint trajectories for joint controllers to track. The control policy outputs each joint’s control law, including stiffness, damping, and feed-forward torque. The joint controllers then execute that control law so that the joint can follow the reference trajectory and stabilize the torso under external disturbances.

### 1.3. Contributions

Our main contributions are as follows:1.We propose a new control framework for quadruped locomotion, which uses a low-frequency RL-based policy to provide a control law allowing for high-frequency model-based joint controllers to resist disturbances.2.We realized high-speed locomotion on challenging terrains with a relatively low computational cost. Our method succeeds in controlling the quadruped to traverse uneven terrains at the speed of 1.8 m/s and frequency of 50 Hz, with a latency of 20 ms.

The rest of this paper is arranged as follows. Section 2 will describe the control framework and the control policy’s training progress in detail. Section 3 will provide the results of simulation experiments, including the behavior of the robot on multiple terrains and the evaluation of our method’s ability to allow for running on uneven terrains. Section 4 contains conclusions and discussions for future works.

## 2. Methods

### 2.1. Control Framework

Our control framework consists of a state estimator, a reference trajectory generator, an RL-based control policy running at a low frequency, and joint controllers running at a high frequency. The control policy will be described in detail in Section 2.2.

Figure 1 shows our control framework. The red part is the RL-based control policy with an update rate of 50 Hz. The rest of the control framework and the physics simulation update at 1 kHz. A 20 ms latency is applied to the control policy to simulate the computation time of the policy model.

As we employ the adjustable joint control law, the impedance characteristics of joints can be modified to suit different situations during locomotion, such as foot contact or leg swinging. With the control law optimized by the RL algorithm, our framework can achieve a better performance compared with controllers with fixed PD gains.

The reference trajectory of the robot can be used to regulate robot behavior and narrow the exploration space of the RL-based policy. While many prior approaches use dynamics models to generate physically feasible reference trajectories for bipedal locomotion [11,12], using fixed reference trajectories proved to be feasible for quadrupedal locomotion [4] as the control policy is capable of modifying the trajectory to stabilize the robot. In our framework, we use a reference trajectory generator to build trajectory, regardless of the robot’s current state. This is because the robot’s state changes violently during high-speed locomotion, making it unsuitable for generating a stable reference trajectory.

The reference trajectory generator provides a periodic reference joint trajectory based on velocity command $v_{\mathrm{xy},\mathrm{cmd}}$, target torso height $h_{\mathrm{torso},\mathrm{ref}}$, period length *T*, stance phase ratio μ, foot raise height $h_{\mathrm{raise}}$, and foot nominal position $x_{\mathrm{foot},0},y_{\mathrm{foot},0}$. For the three joints $\left[q_{3k-2,\mathrm{ref}},q_{3k-1,\mathrm{ref}},q_{3k,\mathrm{ref}}\right]$ of leg *k*, their reference position can be expressed as a function of phase $\phi_{k}$:(1)$${\left[q_{3k-2,\mathrm{ref}},q_{3k-1,\mathrm{ref}},q_{3k,\mathrm{ref}}\right]}^{T}={JTG}_{k}\left(\phi_{k}\right)$$
where $JTG_{k}$ is the joint trajectory generator for leg *k*; phase $\phi_{k}$ is a periodic function of time *t*: $\phi_{k}=\left(t/T+\phi_{k,0}\right)\;\mathrm{mod}\;1$. $\phi_{k,0}$ represents the phase offset for leg *k*’s motion. In this paper, we used trotting gait as a reference trajectory. The phase offset for FR, FL, RR, and RL leg (k=1,2,3,4, respectively) was 0,0.5,0.5,0. As the reference trajectory was independent of feedback information, it can be computed offline, reducing the burden of real-time computing. The reference joint velocity was obtained by differentiating the reference joint position.

In this paper, we used manually designed function $JTG_{k}$. The function generates stance phase joint trajectory by computing inverse kinematics (IK) from the foot reference trajectory, which moves backwards in the torso frame. During the beginning and end of the swinging phase, IK is also used to plan lifting and landing trajectory. The function uses joint space interpolation as a reference trajectory in the middle part of the swinging phase. All parameters of the function $JTG_{k}$ were manually designed, with the purpose of generating a joint trajectory for a command velocity up to 3 m/s while keeping joint velocity under 40 rad/s and foot-raise height under 12 cm.

Figure 2 is a sampled foot trajectory in X-Z plane at the command forward velocity of 2 m/s. This foot trajectory is converted from the joint trajectory generated by the function $JTG_{k}$. The maximum joint velocity of the joint trajectory is 21.2 rad/s.

A joint controller tracks reference joint trajectory with the stiffness $k_{p,i}$, damping $k_{d,i}$, and feed-forward torque $\tau_{i,\mathrm{ff}}$ parameters given by the control policy described in the next section. For joint controller *i*, the control law can be expressed as:(2)$$\tau_{i}=k_{p,i}\left(q_{i,\mathrm{ref}}-q_{i,\mathrm{real}}\right)+k_{d,i}\left({\dot{q}}_{i,\mathrm{ref}}-{\dot{q}}_{i,\mathrm{real}}\right)+\tau_{i,\mathrm{ff}}$$With simple control laws, the joint controller can run at a high frequency.

We used an extended Kalman filter (EKF)-based state estimator for the quadruped, similar to Agarwal et al. [22]. As the orientation estimation noise grows too large during highly dynamic locomotion, we used the orientation integration readout from the Inertial Measurement Unit (IMU) as the estimated orientation of the torso state.

### 2.2. Reinforcement Learning Control Policy

#### 2.2.1. Observation Space and Action Space

The control policy model’s input is a 67-dimension vector, including torso state, command, reference joint trajectory, joint tracking error, and two stance phase indicators ${\mathrm{stance}}_{1},{\mathrm{stance}}_{2}$. ${\mathrm{stance}}_{k}$ is the stance phase indicator, which is set to 1 when the reference motion of leg *k* is in stance phase. Since we used a trotting gait where diagonal feet step synchronously, only two indicators are required as input. A detailed description of these inputs is listed in Table 1.

The policy model’s output is a 72D vector, which needs further processing to obtain joints’ control law parameters. The output vector can be divided into 6 parts; each part contains 12 scalars corresponding to 12 joints. The 6 parts are the scaled stiffness, damping, and feed-forward torque parameters for stance phases (${\hat{k}}_{p,\mathrm{stance}},{\hat{k}}_{d,\mathrm{stance}},{\hat{\tau }}_{ff,\mathrm{stance}}\in R^{12}$), and that set of parameters for swing phase (${\hat{k}}_{p,\mathrm{swing}},{\hat{k}}_{d,\mathrm{swing}},{\hat{\tau }}_{ff,\mathrm{swing}}\in R^{12}$), which are limited to the range of [−1, 1]. For joint *i*, belonging to leg *k*, its control parameters can be computed as:(3)$$\left[\begin{array}{c} k_{p,i}\\ k_{d,i}\\ \tau_{i,ff} \end{array}\right]=\left\{\begin{array}{cc} \left[\begin{array}{c} k_{p,0}\exp\left(\alpha_{p}{\hat{k}}_{p,\mathrm{stance},i}\right)\\ k_{d,0}\exp\left(\alpha_{d}{\hat{k}}_{d,\mathrm{stance},i}\right)\\ \tau_{0}{\hat{\tau }}_{ff,\mathrm{stance},} \end{array}\right] & \;if\;{\mathrm{stance}}_{k}=1\\ \left[\begin{array}{c} k_{p,0}\exp\left(\alpha_{p}{\hat{k}}_{p,\mathrm{swing},i}\right)\\ k_{d,0}\exp\left(\alpha_{d}{\hat{k}}_{d,\mathrm{swing},i}\right)\\ \tau_{0}{\hat{\tau }}_{ff,\mathrm{swing},} \end{array}\right] & \;if\;{\mathrm{stance}}_{k}=0 \end{array}\right.$$
In the equation above, $k_{p,0},\;\alpha_{p},\;k_{d,0},\;\alpha_{d},\tau_{0}$ are empirically set parameters. In our experiments, the values of these parameters are set as $k_{p,0}=12\;\mathrm{Nm}/\mathrm{rad}$, $\alpha_{p}=1.5$, $k_{d,0}=1\;\mathrm{Nms}/\mathrm{rad}$, $\alpha_{d}=1$, $\tau_{0}=15\;\mathrm{Nm}$.

This set of parameters allows for a large range of stiffness and damping coefficients, covering the values presented in some prior works [5,8,9,19]. The large range of PD control laws allow for the robot to exhibit different characteristics under disturbances.

In our control framework, the control policy updates much more slowly than the trajectory generator. As a result, when a leg experiences a phase shift between swing and stance phases, the control policy may not be ready to update. As swinging legs and supporting legs have different dynamics characteristics, the optimal control law should be different. To change the control law at the phase shift without updating the control policy, our policy provides two sets of parameters for each joint, for stance and swing phases, respectively. In addition, we used exponential scaling for stiffness and damping parameters, which allows for the policy model to control these two parameters more precisely when the parameters are small. This design is based on an observation that a low-impedance joint’s dynamics characteristics change more than a high-impedance joint when the same amount of modification is applied to their stiffness or damping coefficient.

#### 2.2.2. Policy Training

We used proximal policy optimization [23] for training. Our policy model was implemented by a fully connected neuron network of five layers. The sizes of the four hidden layers are 256, 256, 128, and 128. The output of the last hidden layer was decoded to obtain the mean and variance of the action, and the estimated state value. We used exponential linear units (ELU) [24] as activation functions in the neuron network model; we applied a running mean to the input and inverse running mean to the estimated state-value output to boost training speed. The sampled action was clipped into the range of [−1, 1]. The training algorithm was implemented in the rl_games [25] library.

#### 2.2.3. Training Environment and Domain Randomization

We implemented our training environment using a Mujoco dynamics engine [26]. We set up an environment with multiple obstacles of 6cm in height and a terrain with a changeable surface. To speed up the process of data collection, we maintained a group of independently running environments and updated them simultaneously with parallel computing. Figure 3 shows one of the training environments. The grey object is the terrain and the red ones are obstacles with a fixed position and shape.

We implemented the reference trajectory generator and joint controller described above in the training environment (including the latency of policy updates). They updated together with the physics simulator. A state estimator was not necessary as the state of the robot can be acquired via the simulator’s API. As our control policy updates at a low frequency, the physics simulator updates multiple times in one environment update step. Here, we used $N_{\mathrm{sim}}$ to denote the number of physics simulator updates in one environment step. For a variable *X*, we defined $X_{\mathrm{sub}\left(j\right)}$ to be a value of *X* at the jth physics simulator update.

We applied domain randomization to increase the robustness of our control policy. We randomize robot dynamics and environment parameters. All randomization was carried out at the beginning of an episode. The randomized parameters are listed in Table 2.

The gravity direction is more heavily randomized around the Y axis (changes more along the X axis), as we hope that the robot will have the ability to run up and down slopes instead of resisting lateral tilts.

#### 2.2.4. Reward Function Design

Our reward function is defined as:(4)$$r=\max\left(\begin{array}{c} w_{\mathrm{lin}}r_{\mathrm{lin}}+w_{\mathrm{rot}}r_{\mathrm{rot}}+w_{\mathrm{yaw}}r_{\mathrm{yaw}}+w_{\mathrm{ori}}r_{\mathrm{ori}}+w_{h}r_{h}\\ +w_{\mathrm{jpos}}r_{\mathrm{jpos}}+w_{\mathrm{torq}}r_{\mathrm{torq}}+w_{\mathrm{jerk}}r_{\mathrm{jerk}}+w_{\mathrm{vjerk}}r_{\mathrm{vjerk}} \end{array},0\right)$$The detailed formula of each term is listed in Table 3. The terms are categorised into five groups: task, stability, smoothness, tracking and limit. Task-related reward terms include $r_{\mathrm{lin}},\;r_{\mathrm{rot}},\;r_{\mathrm{yaw}}$, which encourage the robot to follow the velocity and yaw command. Stability reward terms include $r_{\mathrm{ori}},\;r_{h},\;r_{\mathrm{torq}}$, which keep the robot from falling or making violent moves. Smoothness reward term $r_{\mathrm{torq}}$ penalizes the robot for violent moves. The task, stability and smoothness terms are inspired by the reward settings of Margolis et al. [8]. The tracking term $r_{\mathrm{jpos}}$ encourages the joints to follow the reference trajectory, similar to the reward term in the work of Xie et al. [17]. jerk and vjerk are limit terms, which penalize the joints when moving to infeasible positions or moving faster than their mechanical limit during high-speed motion.

Parameter $q_{i,\max},\;q_{i,\min},\;{\dot{q}}_{i,\max}$ poses a soft limit on joint position and velocity. $r_{\mathrm{jerk}}$ is set to 1 if any of the joints reach the soft position limit; otherwise, it is set to 0. Similarly, $r_{\mathrm{vjerk}}$ is set to 1 if any of the joints reaches the soft velocity limit. The episode terminates if the robot touches the ground with parts other than foot or shank.

#### 2.2.5. Continuous Adaptive Curriculum Learning

Curriculum learning has been shown to be able to improve the performance of the policy, including the success rate when traversing uneven terrains [4] and the maximum speed of the robot [8].

We present a new adaptive curriculum, which gradually tunes two parameters: the terrain height and forward speed command. During training, we used three types of terrain whose difficulty is relative to the terrain’s height. The three types of terrain are shown in Figure 4. Different from some other methods [8], our curriculum changes each environment’s difficulty independently, without the need to compute the overall difficulty distribution.

Similar to Lee et al. [4], we designed a score $p_{j}$ to evaluate the policy’s performance in the jth environment. When an episode ends, $p_{j}$ is determined by the moving average of the robot’s forward velocity $\bar{v}$ and the length of the episode:(5)$$p_{j}=\left\{\begin{array}{cc} 1 & \mathrm{if}\;\mathrm{episode}\_\mathrm{length}>\max\_\mathrm{episode}\_\mathrm{length}*0.9\;\mathrm{and}\;\bar{v}>v_{\mathrm{cmd},j}-0.2\\ -1 & \mathrm{if}\;\mathrm{episode}\_\mathrm{length}<\max\_\mathrm{episode}\_\mathrm{length}*0.5\;\mathrm{or}\;\bar{v}<v_{\mathrm{cmd},j}-0.5\\ 0 & otherwise \end{array}\right.$$

For the *j*th environment, we used $v_{\max,j}$ and $h_{\max,j}$ to denote its maximum difficulty. Its terrain height $h_{j}$ and forward velocity $v_{\mathrm{cmd},j}$ are updated according to the following curriculum at the end of an episode:(6)$$\begin{array}{cc} v_{\mathrm{cmd},j} & \leftarrow \max\left(\min\left(v_{\mathrm{cmd},j}+U\left[-0.1,0.1\right]+p_{j}*0.3,v_{\max,j}\right),0\right)\\ h_{j} & \leftarrow \max\left(\min\left(h_{j}+U\left[-0.005,0.005\right]+p_{j}*0.01,h_{\max,j}\right),0\right) \end{array}$$

The maximum difficulty parameters were assigned to each environment at the beginning of the training. During training, these parameters limit the distribution of difficulty. This limitation keeps a portion of easy environments, which prevents the policy from overfitting with the hard evironments while behaving poorly in easy ones. The way we set the maximum difficulty parameters is shown in Algorithm 1.
**Algorithm 1:** Assigning maximum environment difficulty
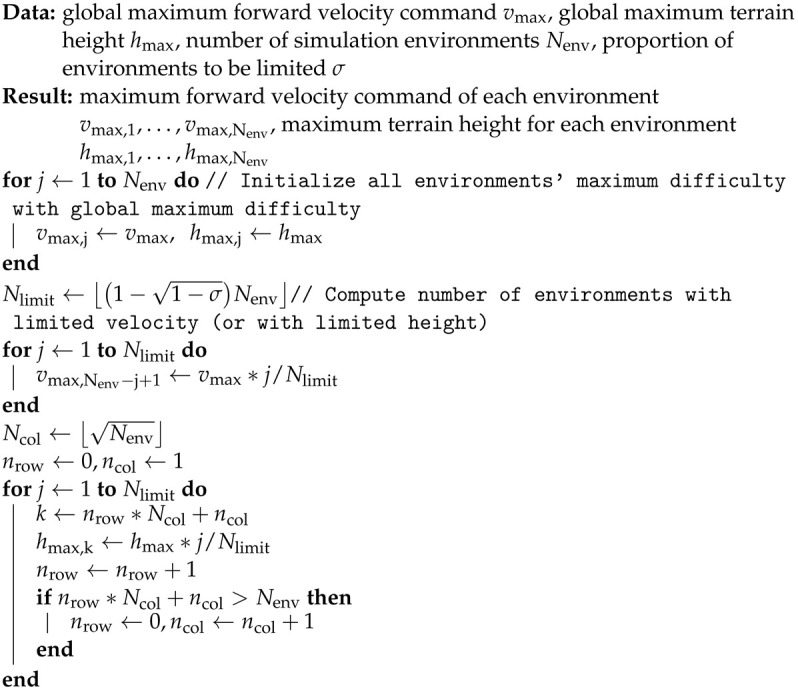


## 3. Results

### 3.1. Training

We trained our policy on a desktop PC with 32-core Intel Xeon Gold 6234 CPU and Nvidia GeForce RTX 2070 GPU. A total of 1024 simulations were computed in parallel by the CPU, with 63 threads for data generation. The training took 9.5 h, corresponding to 58 days of simulation. The distribution of curriculum difficulty was examined at the end of the training progress. During the training, 64% of the environments had limited difficulty. As shown in Figure 5, most of the environments reached a relatively high level of difficulty, yet a small portion of them remained easy due to the limitations.

### 3.2. Results of High-Speed Locomotion

We evaluated our control framework in Webots [27] 2021a simulation. In the experiment, we built a series of obstacles and terrains that can be found in the real world, including stairs ( 6 cm per step, 30 cm width), obstacles ( 6 cm height, 10 cm width), lateral stairs ( 6 cm height difference), and an uneven terrain with random height (maximum height 6 cm, horizontal sampling interval 4 cm). We also designed some dynamic terrains, including two seesaws that can rotate when stepped on, and a floating blocks that will sink when stepped on (each block weighs 2 kg, connected to the fixed environment by a vertical slider joint with a stiffness of 2000 N/m and damping of 50 Ns/m). These terrains are lined up in a row so that the robot can run past them in a single experiment.

We use a simulated Unitree A1 robot [28] to conduct all the experiments, as shown in Figure 6. Appendix A contains a full record of running on different terrains. The Unitree A1 robot is a 12 kg small quadrupedal robot with a standing height of 30 cm. During simulation, we limited the torque output of each joint to match the capacity of the real robot (20 Nm for hip joints; 55 Nm for others). However, the noise and latency of the sensors and actuators were not modeled.

During all experiments in this paper, the update rate of the control policy was set to 50 Hz, while the other parts of the control framework updated at 1000 Hz. The simulation timestep was 1 ms. The parameters for the reference trajectory were set as follows: $h_{\mathrm{torso},\mathrm{ref}}$ = 0.3 m$,\;T=0.25\;s,\mu =0.3,\;h_{\mathrm{foot}}=12\;\mathrm{cm},\;x_{\mathrm{foot},0}=0\;m,\;y_{\mathrm{foot},0}=0.08\;m$. The lateral velocity and rotational velocity command were set to zero. The command forward velocity was 2.2 m/s when the robot traversed the environment. With a policy delay of 20 ms, the robot can achieve a speed of over 1.8 m/s when running across different terrains while only using proprioceptive sensors. The forward velocity during the experiment is shown in Figure 7. Although the robot encountered ground disturbance that drastically slowed it down, it maintained stability and recovered to normal speed in a short time.

During the experiment, the robot experienced slipping foot, blocked swinging foot, and delayed contact. The policy succeeded in stabilizing the robot in these situations, as shown in Figure 8, Figure 9 and Figure 10.

### 3.3. Locomotion Performance on Uneven Terrain

We evaluated our method’s performance when running on an uneven terrain with random height. The result is compared with a baseline, which is a policy with PD parameters fixed to 28 Nm/rad and 0.7 Nms/rad, respectively. During this experiment, we trained two versions of the policy for each method. The first version lacks a policy update latency and the second one is trained and evaluated with 20 ms of latency.

Both methods have a high success rate when the commanded velocity is lower than 2.4 m/s and the terrain height is less than 0.08 m. However, as the velocity or terrain height increases, our method outperforms the baseline, as shown in Figure 11. Also, our method is less affected by control latency.

When running on a 6 cm high uneven terrain, both methods have a similar performance in terms of velocity-tracking when the command velocity is lower than 2.5 m/s. However, as the command velocity increases, our method can track the command with less variance and fewer tracking errors, as shown in Figure 12. The baseline experienced a degradation in performance when there was a latency of 20 ms, while our method was less affected.

The better performance obtained by our framework under the presence of latency may be due to the utilization of the joint controllers running at a high frequency. Since the joint controllers are not affected by the latency, they can react to disturbances in time. With a properly set controller law, these controllers can keep the robot from falling until the policy can respond to the disturbance.

## 4. Conclusions and Discussion

We presented a control framework containing an RL-based control policy with a low update rate and model-based joint controllers with a high update rate. This control framework can achieve high-speed locomotion on uneven terrains. With a latency in policy updates, our control policy stabilized the robot following various disturbances by modifying joint control laws. By providing parameters for a control law, a high-level controller can define the behavior of joints and how they respond to disturbances, as long as the control law is simple enough to update at a high frequency. Since our framework can withstand latency and does not require in-time updates, it can be deployed on robot controllers with an inferior computational capacity.

Our method has the following points to be improved.

1.We only used proprioceptive sensors, which could be combined with exteroception in the future for robots running over high obstacles, as was conducted in Miki et al. [18] and Zhuang et al. [29].2.The manually designed reference trajectory could be further optimized for higher speed.3.The linear feedback control laws could be replaced by non-linear control laws in future works.4.The stability of the controller under sensor noise is not analysed. The stability of a legged controller can be investigated via Poincare Map function [30], or by entropy-based analysis [9].5.Our controller’s robustness under sensor or actuator faults could be investigated in future works. The sensor fault could be detected [31] and the RL-based policy is proven to be able to adapt to actuator failures [8].6.The range of joint controller stiffnesses and damping parameters was empirically set. How the range of these parameters affects the performance of our controller may be investigated in future works.

## Figures and Tables

**Figure 1 biomimetics-09-00037-f001:**
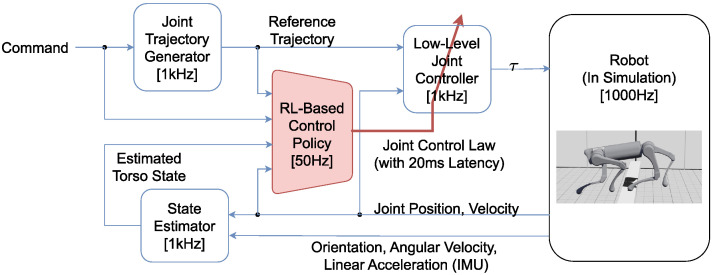
The proposed control framework. The input of the control policy is sampled at 50 Hz.

**Figure 2 biomimetics-09-00037-f002:**
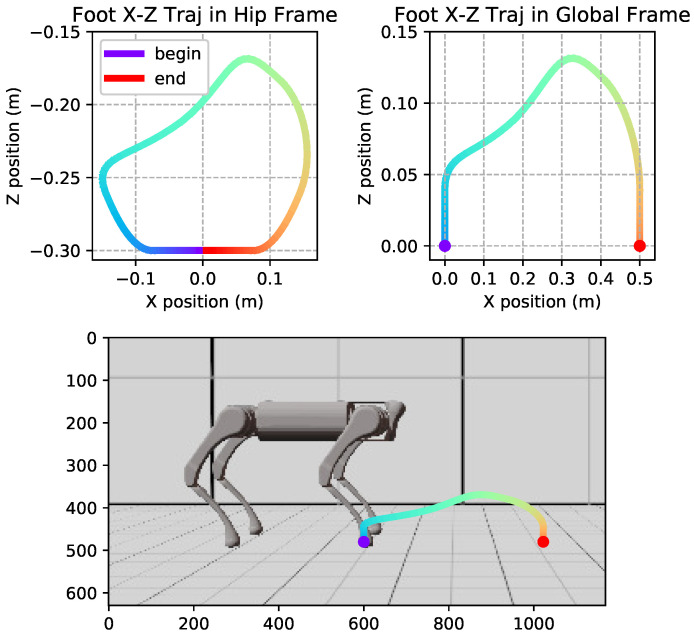
**Top Left:** Reference foot trajectory in hip frame. Hip frame is a frame fixed to torso frame. Its origin is at the hip joint, while its axis is parallel to the axis of torso frame. **Top Right:** Reference foot trajectory relative to the ground. **Bottom:** The scale of reference foot trajectory in the global frame, compared with the robot.

**Figure 3 biomimetics-09-00037-f003:**
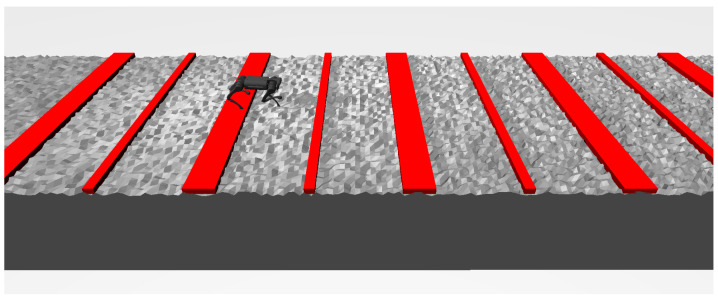
Training environment.

**Figure 4 biomimetics-09-00037-f004:**
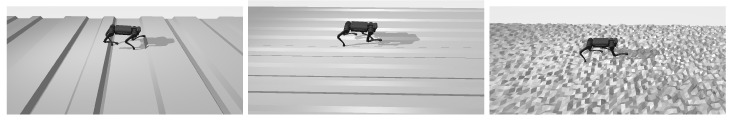
Three types of terrain in training environment: **left:** stairs; **middle:** lateral stairs; **right:** uneven terrain with random height. The obstacles are not displayed.

**Figure 5 biomimetics-09-00037-f005:**
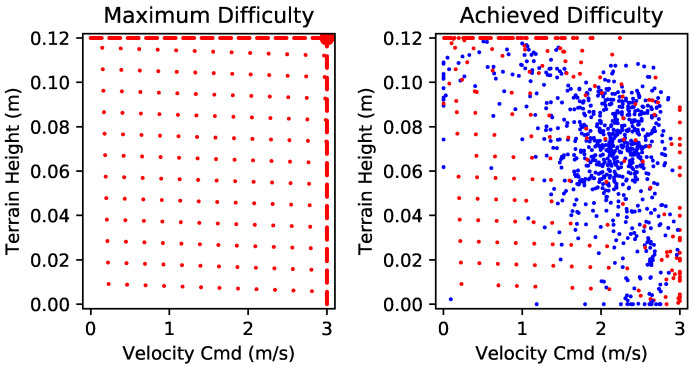
Environments’ difficulty distribution. **Left:** Maximum difficulty distribution. The large dot in the top-right corner indicates all unlimited environments. **Right:** The distribution of environment difficulty at the end of the training. Those who reached at least one of the limits are colored red; the others are colored blue.

**Figure 6 biomimetics-09-00037-f006:**
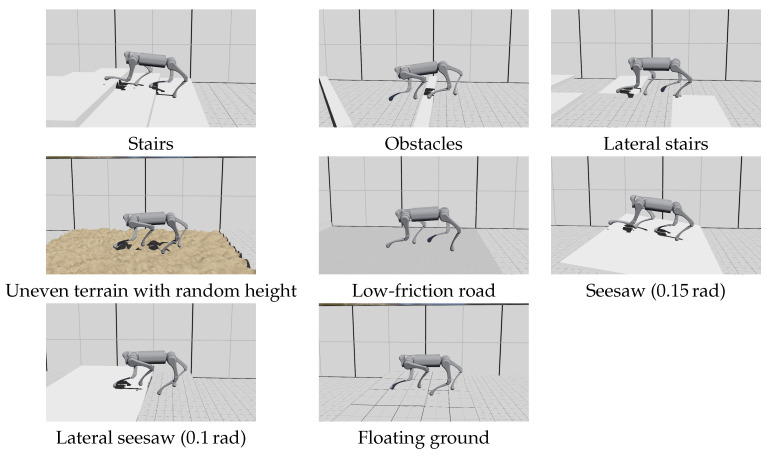
Robot running on different types of terrain with only proprioceptive sensor inputs.

**Figure 7 biomimetics-09-00037-f007:**
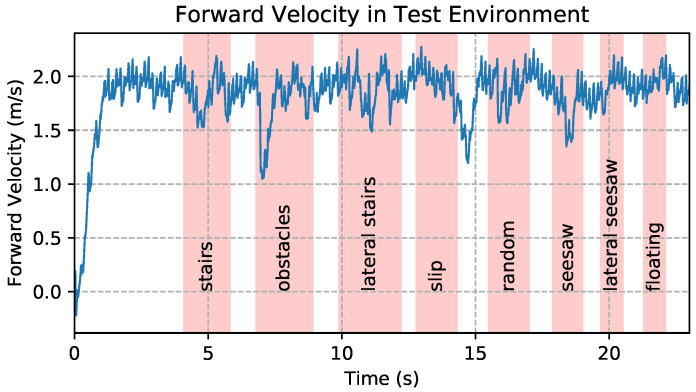
Robot forward velocity during the experiment.

**Figure 8 biomimetics-09-00037-f008:**
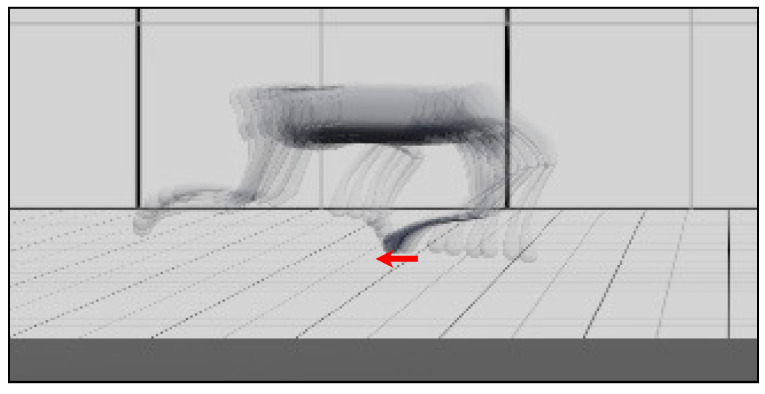
The rear-right leg slips on slippery ground (with the friction coefficient of 0.15). Its motion is indicated by the red arrow. Multiple snapshots of the motion in 0.05 s are stacked together to visualize the slipping. The duration of the slipping is about 66% of the stance phase.

**Figure 9 biomimetics-09-00037-f009:**
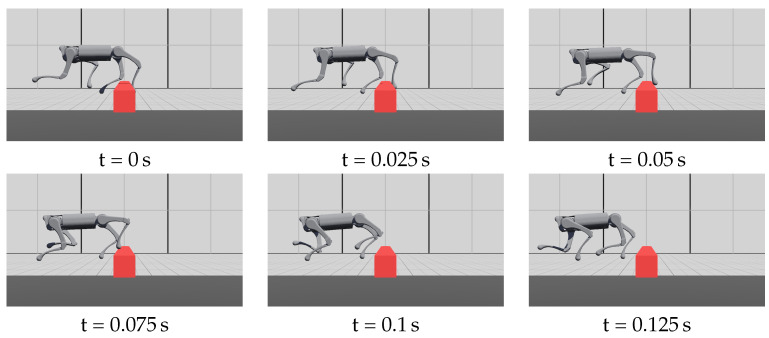
The rear-left foot is struck by an obstacle. The robot managed to raise the blocked leg over the obstacle. The torso state was not significantly disturbed and the robot could keep running. Appendix A records multiple successful cases under the disturbance of an obstacle.

**Figure 10 biomimetics-09-00037-f010:**
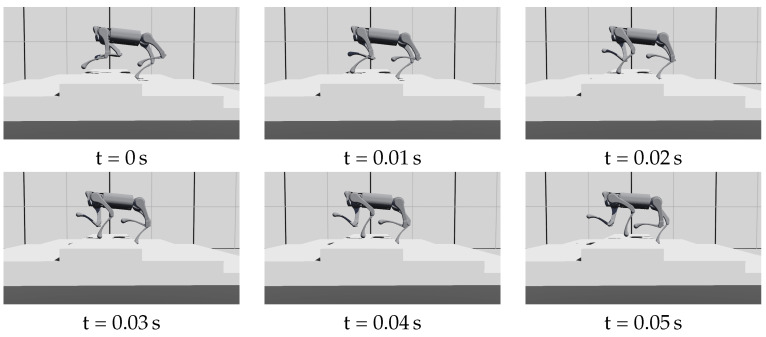
The front-left foot makes contact with the ground after more than half of the stance phase (at T = 0.02 s), then leaves the ground (at T = 0.04 s). This stance phase is less than 0.02 s (27% of planned stance phase). Appendix A records the complete motion of the delayed contact.

**Figure 11 biomimetics-09-00037-f011:**
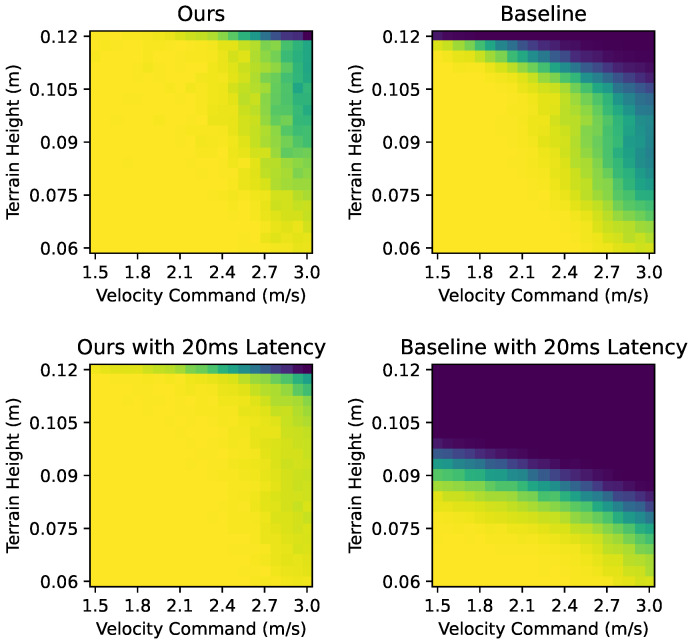
Success rate when running on an uneven terrain with random height.

**Figure 12 biomimetics-09-00037-f012:**
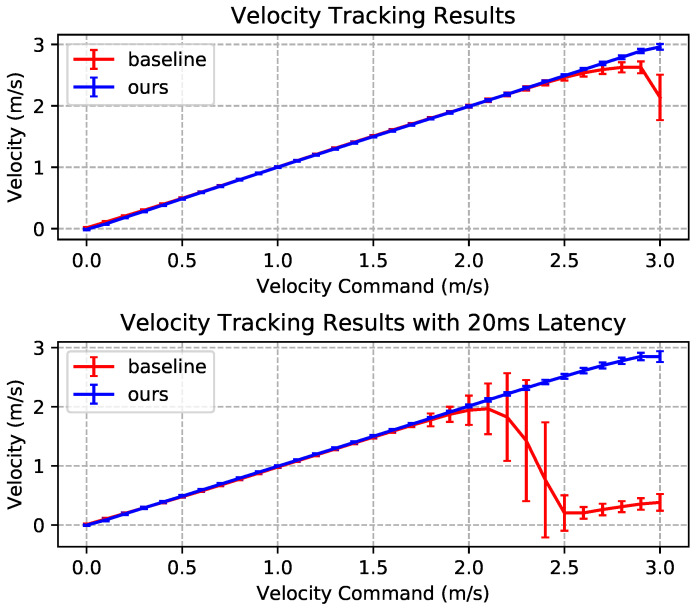
Achieved velocity against command velocity when running on uneven terrain with a height of 0.06 m.

**Table 1 biomimetics-09-00037-t001:** List of control policy inputs.

	Observation	Size
Torso state	torso linear velocity $v_{\mathrm{torso}}$	3
	torso angular velocity $\omega_{\mathrm{torso}}$	3
	gravity direction	3
Commands	linear velocity command $v_{\mathrm{xy},\mathrm{cmd}}$	2
	angular velocity command $\omega_{z}$	1
	yaw difference ^1^	2
	gait period *T*	1
	stance phase ratio μ	1
	ref. torso height $h_{\mathrm{torso},\mathrm{ref}}$	1
Reference traj	ref. joint position $q_{\mathrm{ref}}$	12
	ref. joint velocity ${\dot{q}}_{\mathrm{ref}}$	12
Tracking error	joint position error $q_{\mathrm{ref}}-q_{\mathrm{real}}$	12
	joint velocity error ${\dot{q}}_{\mathrm{ref}}-{\dot{q}}_{\mathrm{real}}$	12
	${\mathrm{stance}}_{1},{\mathrm{stance}}_{2}$	2

^1^ Expressed with $sin\left(\Delta yaw\right)$ and $cos\left(\Delta yaw\right)$.

**Table 2 biomimetics-09-00037-t002:** Domain randomization applied to the environments.

Parameters	Randomization Approach
Ground friction	Set to $U\left[0.3,0.7\right]$
Gravity direction	Rotate by U[−0.1,0.1] around X axis and U[−0.2,0.2] around Y axis
Body mass	Scaled by $U\left[0.6,1.4\right]$

**Table 3 biomimetics-09-00037-t003:** Elements of reward function.

Reward Term	Weight	Expression
$r_{\mathrm{lin}}$	3	$e^{-{∥v_{xy,\mathrm{cmd}}-v_{xy,\mathrm{real}}∥}^{2}/0.02}$
$r_{\mathrm{rot}}$	1	$e^{-{\left(\omega_{z,\mathrm{cmd}}-\omega_{z,\mathrm{real}}\right)}^{2}/0.01}$
$r_{\mathrm{yaw}}$	2	$e^{-{\left(yaw_{\mathrm{cmd}}-yaw_{\mathrm{real}}\right)}^{2}/0.01}$
$r_{\mathrm{ori}}$	2	$e^{-\left(roll_{\mathrm{real}}^{2}+pitch_{\mathrm{real}}^{2}\right)/0.02}$
$r_{h}$	0.5	$e^{-{\left(h_{\mathrm{torso},\mathrm{ref}}-h_{\mathrm{torso},\mathrm{real}}\right)}^{2}/0.05}$
$r_{\mathrm{jpos}}$	0.5	$e^{-\sum_{i}{\left(q_{i,\mathrm{ref}}-q_{i,\mathrm{real}}\right)}^{2}/0.05}$
$r_{\mathrm{torq}}$	−0.001	$\sum_{i}\sum_{j=1}^{N_{\mathrm{sim}}}\tau_{i,\mathrm{sub}\left(j\right)}^{2}/N_{\mathrm{sim}}$
$r_{\mathrm{jerk}}$	−0.5	$1\left\{\sum_{j=1}^{N_{\mathrm{sim}}}1\left\{q_{i,\mathrm{sub}\left(j\right)}>q_{i,\max}\;or\;q_{i,\mathrm{sub}\left(j\right)}<q_{i,\min}\right\}>0\right\}$
$r_{\mathrm{vjerk}}$	−0.5	$1\left\{\sum_{j=1}^{N_{\mathrm{sim}}}1\left\{\left|{\dot{q}}_{i,\mathrm{sub}\left(j\right)}\right|>{\dot{q}}_{i,\max}\right\}>0\right\}$

## Data Availability

The code we used for training the model in this study can be found here: https://github.com/sqrt81/variant_impedance_tracking_RL.

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
