# Peer review of "Learning Quadrupedal High-Speed Running on Uneven Terrain"

_biomimetics, 2024, doi:10.3390/biomimetics9010037_

Round 1
Reviewer 1 Report
Comments and Suggestions for Authors
This paper deals with the problem of reinforcement learning of a quadruped robot on uneven terrains. The topic is interesting. The paper shows balance between theorical results and experiments. The paper requires revision to improve it.
1- The title of the paper is not good and should be changed, by showing for example the words quadruped robot,
2- The section 1.2. Motivation should be subdivided into two subsections where the second one is Contributions.
3- The number of references is low and should be expanded by adding more recent references on related problems for biped robots. There are several works, like
- Learning Biped Locomotion, IEEE Robotics & Automation Magazine
- A Hierarchical Framework for Quadruped Omnidirectional Locomotion Based on Reinforcement Learning, IEEE Transactions on Automation Science and Engineering
- Robust biped locomotion using deep reinforcement learning on top of an analytical control approach, Robotics and Autonomous Systems
There are mor interesting results. Authors should add more discussion in the paper and in the introduction.
4- There is no detail on the different elements used for the reward function in Table 3. Note that there are several types of reward functions according to the literature (add references for those used in Table 3). So, you should explain how you have built or select these elements of the reward function.
5- The computation time for the learning should be added and discussed.
6- Figure 12 is difficult to understand it and more details should be added to make these two plots clearer.
7- The whole paper needs a revision to improve English and some weak sentences.
Comments on the Quality of English LanguageShould be improved
Author Response
Please see the attached WORD file for my response. Thank you.

Reviewer 2 Report
Comments and Suggestions for Authors
This contribution presents original ideas in the study and
advances the previous research in this area. The level of the
originality of contribution to the existing knowledge with an
emphasis on the paper’s innovativeness i n both theory
development and methodology used in the study is very high.
This work makes a significant practical contribution and it makes
impact on the research work on the research community.
The quality of arguments, the critical analysis of concepts,
theories and findings, and consistency and coherency of debate
are well addressed in this paper.
The paper has a good writing style in term of accuracy, clarity,
readability, organization, and formatting.
Nevertheless the following issues should be addressed:
In Fig. 1. the proposed control framework. is porposed. Please address even intuitivelly the problem of the stability of this control strategy. In fact, a feedback is resent in the scheme, this implies that the stability must be considered in your analysis to guarantee the repetibility of the results.
- In Fig. 2 the reference trajectories are proposed. Please discuss more in depth the criteria for this choice.
- Please discuss what can change in the presence of disturbance and noise also with the help of the suggested literature here below. In fact, in case of noise or fault the reader could be interested in havng some suggested litteratre.
Concerning the cited literature you can consider the following papers to improve the tutorial aspects of the paper. These papers are published by MDPI journals.
Tsymbal, O.; Mercorelli, P.; Sergiyenko, O. Predicate-Based Model of Problem-Solving for Robotic Actions Planning. Mathematics 2021, 9, 3044. https://doi.org/10.3390/math9233044
Mercorelli, P. A Fault Detection and Data Reconciliation Algorithm in Technical Processes with the Help of Haar Wavelets Packets. Algorithms 2017, 10, 13. https://doi.org/10.3390/a10010013
Reviewer 3 Report
Comments and Suggestions for Authors
The paper suggests an algorithm for the high speed gaint control in uncertain environment. Authors presented and tested an algorithm that successfully tracks reference trajectory and deals with obstacles of a different types.
The fact that the reference trajectory is independent of the high-frequency feedback that is availible can be considered as a flaw due to potential improovement of the overall performance with introducing limited correction of reference trakectory with accordance to the feedback.
There are also 5 empirically set variables introdused in eq. 3 that are lacking of explanation. It is not clear how the incorrection in their setting could affect presented algorithm and what range of this missetting is acceptable. Some comments on their meaning and influence on the algorithm performance could be another improvement of the article.
Overall, the article contains a solid scientific results and can be recommended to publishing after minor revision.
Comments on the Quality of English LanguageMinor proofreading is required ()
Reviewer 4 Report
Comments and Suggestions for Authors
The paper presents a two-level control architecture designed for high-speed motion of quadruped robots on rough terrains. While the lower level implements classic high-frequency control techniques, the upper level based on RL adjusts the lower-level controllers at a slower frequency.
The results are evaluated through simulations in environments with different types of obstacles (steps, rough terrain, slippery areas, etc.).
The article is well-structured, although some clarifications are needed to fully understand certain aspects and provide reproducibility to the experiments used to validate the proposal.
The main limitation of the presented work is that the results have only been validated through simulation, with no reflection or critique regarding the validity of this approach in a real Unitree A1 robot. Therefore, we urge the authors to:
1. Justify the equivalence of the simulated system (robot and terrain) with a real system. Details should be provided on how the robot is modeled, including aspects such as masses, inertias, mass distribution, actuator power capabilities, possible low-level control algorithms (joint control), proprioceptive sensors and their characterization (precision, sampling frequency, uncertainty, possible latency, etc.), dimensions (obstacle dimensions alone are insufficient without considering the robot's dimensions). Regarding the terrain, values such as friction coefficients, obstacle density, maximum slope of the seesaw, obstacle density on uneven terrain with random height should be included.
2. Include a section analyzing the validity of the results and their applicability when implementing the control on the real robot. What are the practical possibilities of implementing the framework on the robot's control system? Should the original hardware architecture of the robot be expanded, and if so, in what way? Can the software architecture, including real-time access to sensor readings and driver writing, support the method? What aspects have not been simulated, and why are they considered irrelevant? Concerning the terrain, how do the prototype scenarios used align with real-world scenarios? Real-world cases with practical application should be presented to demonstrate how their characteristics match those of simulated scenarios.
Here are some specific points to address:
a) Equation (1): While the meaning of the variables qi can be assumed, they should be explicitly stated in the text. Similarly, it should be clarified whether JTGk is a vector with two components (X, Z).
b) Line 91: Provide a more detailed explanation of how the JTGk trajectory is generated.
c) Figure 2: Clarify whether the trajectory in the left figure corresponds to the Torso Frame or the Leg Frame. Including the robot leg in one of the ground support positions in the drawing would be helpful.
d) Equation 2: Explain how the value of tao_1,ff is selected.
e) Table 2: Use an equation editor for the Expression column to ensure that exponentials are correctly represented.
f) Table 2: Explain why the gravity vector rotates differently on the X-axis (0.1, 0.1) than on the Y-axis (0.2, 0.2). Is there a relationship with gravity, and why were these values chosen?
g) Table 3: Address the term r_hvaria, as it appears to be h_torso,real instead of z_torso,real.
h) Table 3: Maintain consistency in how equations are written, including a parenthesis after the exp operator in the r_jpos expression.
i) Table 3: Clarify the meaning of the "I" in the expressions of r_jerck and r_vjerck.
j) Table 3: Provide an explanation for how r_jerk and r_vjerk are evaluated, particularly the latter, which may not be as evident to comprehend as the previous equations.
k) Figure 4: Change the dark blue background as it hinders the visibility of the robot and environment elements. Consider using a different background color (as done in Figure 6).
l) Line 193: Specify the version of Webots used.
m) Figure 6: Provide details about some terrains, including the coefficient of friction for the Low Friction Road, obstacle density, and the distribution of their dimensions (base x height) for Uneven Terrain, the maximum slope of Seesaw terrains, and how the floating ground sinks or varies, whether it does so constantly or depends on the weight it supports (differently when all four legs are in contact compared to other cases).
n) Line 201 and following: Provide detailed information about all aspects included in the simulated model of the A1 in Webots, including its geometric data, mass distribution, included sensors and their models, included actuators, and their constraints (maximum torque that the control law can apply to prevent actuator saturation, for example).
We encourage the authors to address these points to improve the quality and usefulness of their work for other researchers.
Reviewer 5 Report
Comments and Suggestions for Authors
This is a well-written paper with high-quality research. I have the following comments.
1. Line 70: The authors should explain the difference between the proposed method and the existing methods in detail. The authors stated that this new control framework comprises a low-frequency RL policy and high-frequency joint controllers. Based on the author’s review, some existing works (ref 8-11) employed a similar framework. What makes the proposed method better than other existing works?
2. Line 89: the authors manually designed the joint trajectory generator function JTGk. Can the author mathematically express this function? Is such a function specifically designed for the study in this paper? Or can this function be universally applied to similar applications?
3. Line 159: “The meaning of each term is listed in table 3.” Indeed, table 3 lists the mathematical expression of each term in the reward function. It would be better if the authors could add some explanations in Table 3 about the meaning of each term.
4. Line 228: A more in-depth discussion is needed for Figure 12. This is related to comment 1. Which part of the proposed framework improves the performance such that latency has less effect?
5. The authors also stated the proposed method has good computational efficiency. Can the author compare the method’s computation time with the baseline?
6. It would be valuable if the authors could provide videos corresponding to Figures 9 and 10 as supplementary materials.
Comments on the Quality of English LanguageThe English language may need some minor checking/correction, especially the mix of present and past tenses in the abstract.
Round 2
Reviewer 1 Report
Comments and Suggestions for Authors
Paper has been imporved according to the proposed comments.
Reviewer 4 Report
Comments and Suggestions for Authors
The suggestions in my previous review have largely been correctly taken into account. For this reason, I consider that the article can be published in its current revision.